# What is the financial burden to patients of accessing surgical care in Sierra Leone? A cross-sectional survey of catastrophic and impoverishing expenditure

Manraj Phull,[1] Caris E Grimes [iD] ,[2,3] Thaim B Kamara,[4] Haja Wurie,[5] Andy J M Leather,[6] Justine Davies[7,8]

► Prepublication history and additional materials for this paper is available online. To view these files, please visit the journal online (http://dx.doi.org/10.1136/bmjopen-2020-039049).

AJML and JD are joint senior authors.

For numbered affiliations see end of article.

**Correspondence to**
Professor Justine Davies;
j.davies.6@bham.ac.uk

## ABSTRACT

**Objectives** To measure the financial burden associated with accessing surgical care in Sierra Leone.

**Design** A cross-sectional survey conducted with patients at the time of discharge from tertiary-level care. This captured demographics, yearly household expenditure, direct medical, direct non-medical and indirect costs for surgical care, and summary household assets. Missing data were imputed.

**Setting** The main tertiary-level hospital in Freetown, Sierra Leone.

**Participants** 335 surgical patients under the care of the hospital surgical team receiving operative or non-operative surgical care on the surgical wards.

**Outcome measures** Rates of catastrophic expenditure (a cost >10% of annual expenditure), impoverishment (being pushed into, or further into, poverty as a result of surgical care costs), amount of out-of-pocket (OOP) costs and means used to meet these costs were derived.

**Results** Of 335 patients interviewed, 39% were female and 80% were urban dwellers. Median yearly household expenditure was US$3569. Mean OOP costs were US$243, of which a mean of US$24 (10%) was spent prehospital. Of costs incurred during the hospital admission, direct medical costs were US$138 (63%) and US$34 (16%) were direct non-medical costs. US$46 (21%) were indirect costs. Catastrophic expenditure affected 18% of those interviewed. Concerning impoverishment, 45% of patients were already below the national poverty line prior to admission, and 9% of those who were not were pushed below the poverty line following payment for surgical care. 84% of patients used household savings to meet OOP costs. Only 2% (six patients) had health insurance.

**Conclusion** Obtaining surgical care has substantial economic impacts on households that pushes them into poverty or further into poverty. The much-needed scaling up of surgical care needs to be accompanied by financial risk protection.

## INTRODUCTION

An estimated 33 million individuals globally face financial catastrophe through payment for surgery and anaesthetic care each year.

## Strengths and limitations of this study

► Use of exit interviews to provide in-depth data on costs of accessing surgical care.
► Thorough and detailed capture of household expenditure.
► Provides reliable estimates of out-of-pocket, catastrophic and impoverishing expenditure as well as sources of financing.
► Data captured in one hospital only, although that is the major surgical care centre for the country.
► Only examines those who accessed care and does not allow exploration of costs as a limitation to accessing care.

Furthermore 3·7 billion people have been estimated to be at risk of catastrophic expenditure (CE; defined as a total out-of-pocket (OOP) health payment that exceeds a set threshold of the household's annual income or expenditure) due to a lack of financial risk protection (FRP).[1,2] Surgical conditions make up 30% of the global burden of disease, and globally, an additional 143 million surgical procedures are required annually to meet the current unmet surgical need.[1,3] To ensure universal health coverage, it is therefore essential that FRP is prioritised alongside the scaling up of surgical care. The Lancet Commission on Global Surgery stated a target of 100% financial protection by 2030 for people accessing surgical care, and FRP indicators for surgery are now included within the World Development Indicators.[4] Despite this, there is little information on financial implications of accessing surgery in the literature beyond modelled studies,[1,2,5] many of which have been based on few real-world data points.

BMJ

Worldwide modelled data on CE and impoverishing expenditure (IE; defined as being pushed into or further into poverty) related to surgical care reveals that those most affected are individuals in low-income and middle-income countries (LMICs).[1 2 6] Modelling studies from Sierra Leone, classed as 'least developed' by the UN and with a population of 7 million, reflect these findings; between 84.7% and 49.9% of the population in Sierra Leone is estimated to be at risk of CE if they require surgery. Estimated average OOP costs for major surgery in the country were US$117.60, which put 59.2% to 73.3% of the population at risk of impoverishment.[5 7] However, there are no empirical data to validate these estimates. The estimated unmet surgical burden of disease in Sierra Leone is huge, at 92%, as a result of the historical neglect of surgical care both nationally and globally.[8–10] To enable effective planning of surgical services in future, an accurate understanding of the financial implications of accessing surgical services is required.

In Sierra Leone, as in many LMICs, payments for healthcare are upfront, complex and not immediately apparent from hospital-listed service charges. In addition, hospital-listed charges—where they exist—may not reflect the total facility-incurred costs that patients pay during their hospitalisation. These include direct medical costs that are charges for the payment of medical care and direct non-medical costs that include items such as transport to the hospital and food. In addition, substantial costs of care may be incurred prior to the hospitalisation episode. For example, there may be direct medical costs at other healthcare facilities visited prior to the definitive admission. Finally, there are indirect costs (eg, loss of wages while receiving care) that patients, and in some cases their caregivers, experience in their illness, which also impact on ability to access care. Two ways of capturing these costs is the measurement of IE or CE. The two most widely used thresholds for CE are an expense of >10% of total annual expenditure or >40% of non-subsistence expenditure (ie, household expenditure net of subsistence costs, as a means of capturing the ability to pay).[11–14]

This study aimed to measure the financial burden associated with receiving surgical care in Sierra Leone by using an exit survey to determine: (A) direct medical, direct non-medical and indirect OOP costs to pay for a surgical care episode, (B) the rate of impoverishment and CE, (C) the wealth characteristics of the population accessing surgical care relative to that of the general Sierra Leonean population, (D) the factors associated with higher costs of hospital care, (E) the in-hospital payment mechanism (ie, where and to whom the OOP payments are being made) and (F) how costs of accessing surgical care are met and the factors associated with meeting costs of care.

## METHODS
### Setting
This study was done in the main tertiary referral centre in Sierra Leone, located in the central part of greater Freetown, and where the majority of surgical care in the countries' non charitable sector is done. It is a 400-bed hospital with 150 beds dedicated to surgical care. Surgical care is delivered in 5 of the 10 wards, an accident and emergency department with a trauma ward for short stay (<24 hours) emergency surgical patients, a surgical outpatient unit, an intensive care unit and five operating theatres. The average surgical volume is 80–100 operations per month.[15] The surgical department is run by eight surgical and two anaesthetic consultants covering six specialities: general surgery, surgical oncology, urology, paediatrics, trauma and orthopaedics, and ear, nose and throat (ENT) surgery. Obstetric and gynaecological surgical care is delivered at a nearby tertiary referral hospital dedicated to women's health, where all pregnant and lactating women receive free healthcare under the government's free healthcare initiative and therefore not included in this study.

### Participants
Participants were all surgical patients who consented to take part, receiving operative or non-operative surgical care under the care of the hospital surgical team and located on one of the surgical wards. Patients under the care of non-surgical teams; patients under the age of 16 years who were without a parent, guardian or head of the household; and participants unable to consent and/or unwilling to take part in the study were excluded. Participants were recruited consecutively to the study on admission for surgical care from June to August 2018.

### Data collection
A structured questionnaire was administered to patients and/or their relatives at the time of formal discharge from surgical care while patients were on the ward. Where patients self-discharged or left against medical advice, where possible they were interviewed when leaving the hospital. Interviews were conducted in a private space, and all participants were encouraged to bring a relative, head of the household or the main breadwinner to allow for expenditure and OOP costs to be captured accurately.

The questionnaire was designed based on tools used in similar studies done in LMIC settings.[16–19] It was co-designed with in-country experts, healthcare professionals and researchers to ensure that the questions were suitable for the Sierra Leone context. The questionnaire was pilot-tested for ease of comprehension, clarity of definitions, appropriateness of questions and manageability of the length of the interview in six patients (who were excluded from the analysis). Minor modifications were made to the wording of the questions based on this, but the meaning of the questions was not changed. The questionnaire was designed and written in English and administered by trained Sierra Leonean research assistants in either English or a chosen local dialect (most commonly Krio). Data were captured on paper and later transferred to electronic format.

## Definition and construction of variables

Data were collected on the particpants' age, gender and address (later used to determine if they were resident in an urban or rural area). The occupation of the main breadwinner was recorded using free text followed by a question on whether this was salaried (ie, employed) or non-salaried (ie, self-employed or working in the informal sector). Education was captured as the highest level of education of the main breadwinner. Information on household expenditure was captured by asking 7 questions on regular items purchased in a typical week (food, drink and so on), 11 questions on larger expenditure items typically purchased monthly (toiletries, clothing and so on) and a further 12 questions on typical yearly spend on big household items such as furniture and livestock (see online supplemental appendix 1). Total food expenditure ($foodexp_h$) was summed as a separate variable for the purposes of calculating CE (where food expenditure was used to define subsistence costs). Number of people living in the household ($HHsize$) was also captured, as was the number of days of sickness before presentation, whether care had been sought elsewhere prior to presentation at Connaught Hospital and the mode of transport used.

Data were also collected on the following: whether the patient was an emergency or elective case; whether the participant was eligible for free healthcare (for patients under the age of 5 years old, pregnant or lactating mothers, Ebola survivors, destitute or disabled patients); and the primary diagnosis, recorded from review of the patient's admission notes, ward and theatre ledgers (later summarised into 10 categories of surgical conditions: trauma, hernia, abdominal conditions, peripheral vascular disease or diabetic foot disease, urological conditions, breast mass/cancer, burns, ENT/dental disease, thyroid, congenital abnormality or paediatrics). Treatment was categorised as operative or non-operative following review of the patient's admission notes. Length of hospital stay was also calculated.

Direct medical OOP costs were captured across the entire illness episode including in-hospital costs (from the point of admission to discharge from the tertiary care hospital) and prehospital costs (for other medical costs related to the admission episode that occurred prior to the tertiary care admission). In-hospital direct medical costs were the sum of administrative costs (including registration, admission, triage, bed and discharge fees), medications, medical supplies, investigations, blood transfusion, operation cost and informal payments (defined as any payment that was not part of hospital policy, such as doctors' fees, tips, payments made to porters and to nursing staff for nursing care). If costs were 'formal', we asked whether these costs were paid directly to the hospital bank/cashiers directly or via hospital staff or to an external facility (such as external pharmacy or laboratory). For prehospital care, non-medical direct costs were calculated from transport costs. For the hospital episode, non-medical direct costs were captured as: cost of transport to the hospital or to and from the hospital to get food, medical supplies and investigations from external facilities and the cost of food and accommodation during the hospital stay. Finally, indirect costs were captured by estimating lost wages during the illness episode.

All costs are presented in Le and US\$ at the conversion rate of 15 July 2019 (1 Sierra Leonean Leone=US\$0.00011567).

Total household expenditure ($totalexp_h$) was calculated over the course of 12 months by summing all the variables collected on all regular household items purchased as described above.

Total OOP payments for surgical care ($OOP_t$):=total direct medical costs+total direct non-medical costs+total indirect costs

CE is most widely defined as either an expense more than 10% of total annual expenditure or an expense of more than 40% of non-subsistence expenditure (ie, household expenditure net of subsistence (here, food ($foodexp_h$)) costs). We considered 10% of total household expenditure to be our main outcome of CE but present results from the 40% of non-subsistence expenditure as a sensitivity analysis.

CE was therefore present if: $\frac{OOP_t}{totalexp_h} > 0.1$

In the sensitivity analysis, using the threshold of 40% of non-subsistence expenditure, CE was present if: $\frac{OOP_t}{totalexp_h - foodexp_h} > 0.4$

IE is defined as being pushed into or further into poverty. The Sierra Leone national poverty line (spending <US\$1.25/person/day) threshold was used for the main analysis. In addition, two further thresholds for poverty were used based on World Bank definitions: 'poverty': spending <US\$3.10/person/day and 'extreme poverty': spending <US\$1.90/person/day.[4] Presence of poverty before (baseline) and after OOP spending on surgical care were then calculated.

Baseline poverty ($BLP_h$) at each threshold was determined to be present if total household expenditure ($totalexp_h$) per individual inhabiting each household divided by the number of days in the year was below the poverty threshold chosen, that is: $\frac{\left(\frac{totalexp_h}{HHsize}\right)}{365} \leq poverty\ line$

Impoverishment as a result of surgical care was defined as present if the total household expenditure net the total OOP costs for surgical care ($totalexp_{netsurg}=totalexp_h - OOPt$) per head of household, per day was less than the chosen poverty threshold

ie, IE present if: $\frac{\left(\frac{totalexp_{netsurg}}{HHsize}\right)}{365} \leq poverty\ line$

Both CE and IE are presented as the number and percentage of participants who experienced CE and or IE.

Summary household asset data were collected using a yes or no response to the ownership of the following assets: electricity/light, mobile phone, radio, television, computer, refrigerator, generator, bicycle, motorcycle and car or truck.

## Sample size and power calculation

Sample size was calculated using the University of San Fransisco California (USCF) online calculator.[20] Based on a similar study done in Uganda that estimated CE to be 31%[16] in a free healthcare setting, modelled and World Bank data for Sierra Leone that estimates CE at 84.7% and 49.9%, respectively, and from discussion with academics with in-country knowledge, we estimated that CE would be around 60% of patients admitted for surgical care. The sample size required to capture this with a CI of 55% to 65%, allowing for 10% loss to follow-up was 442 patients.

## Statistical analysis

Statistical analysis was done using SPSS V.25 for windows.

Characteristics of the population seeking care are described. Normally distributed data are presented as mean and SD, otherwise median, IQR and range are used. Multiple imputation chained equations were used to compute missing data points using predictive mean matching for variables with less than 20% missingness and where missingness was identified as not at random. Where imputed variables were used, the pooled mean is shown as standard SPSS output. A complete case analysis was done for variables describing how costs of accessing care were met and the consequences of accessing care.

Wealth characteristics (household asset ownership) of the population accessing surgical care were compared with those in the general population (2015 Census data[21]) using the $\chi^2$ test.

Associations between direct medical in-hospital OOP costs of care and age, sex, type of admission (emergency or elective), operative or non-operative care, type of operative procedure or length of stay were tested using a generalised linear model using a Tweedie function with a power of 1.9.

## Ethical approval

Ethical approval was granted by the Sierra Leone Ethics and Scientific Review Committee and from the King's College London Research Ethics Committee (ref. LRU-17/18–6455). All patients gave written consent to participate where possible and witnessed thumbprints and verbal consent where patients were illiterate. Patients were given information about the study at admission and consented between 4 and 24 hours later after due time was given to consider the study information. Consent was reconfirmed just prior to doing the exit interview.

## RESULTS

Of the initial 416 recruited participants, a total of 335 were interviewed (figure 1). Participant characteristics are presented in table 1. In summary, the mean age of the interviewed patients was 28 (SD 20). Thirty-nine per cent were female and 80% lived in an urban area. Twenty-nine per cent were formally employed with a further 66.9% being employed but without a regular salary—either self-employed or employed within the informal sector. The level of education of the main breadwinner was secondary school in 38%, college/university in 28% and no formal education in 24%. The median household size was 6 (IQR: 4, range: 4–8) with a mean total yearly household expenditure of US$3569 (see online supplemental appendix table 2 for imputed and non-imputed data and online supplemental appendix table 3 for a comparison with expenditure assessed in the Economic and Financial survey in 2014).[22] Sixty-seven per cent of participants had sought care for their illness elsewhere prior to presentation at the tertiary referral hospital. Seventy-two per cent arrived by public transport, and the majority were classed as emergency admissions (72%). The most common reasons for presentation were trauma, hernia or other abdominal conditions. Sixty-eight per cent underwent operative intervention with the remainder being managed by non-operative measures. Median length of stay was 8 days (IQR: 18, range: 3–21).

The total mean cost for the surgical care episode was US$243, of which US$24 (10%) accounted for prehospital direct costs (medical costs were US$21 and non-medical were US$3). Of the in-hospital direct costs (mean US$172), a mean of US$138 (63%) was due to direct medical costs and US$34 (16%) for direct

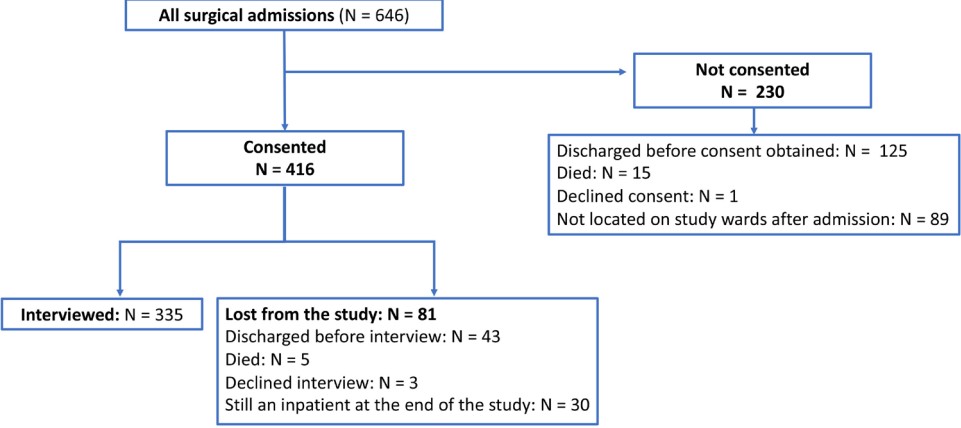

**Figure 1** Study recruitment process diagram.

## Table 1 Participant characteristics

| Demographics of participants | |
|---|---|
| Total number of patients interviewed | 335 |
| Mean age in years (SD) | 28 (20) |
| Female, n (%) | 132 (39) |
| Urban residents, n (%) | 269 (80) |
| Type of job, n (%) | |
| Self employed/Informal sector | 224 (67) |
| Employed | 97 (29) |
| Unemployed/retired | 12 (4) |
| Missing/don't know | 2 (1) |
| Level of education of main breadwinner, n (%) | |
| No formal education | 79 (24) |
| Primary school | 25 (8) |
| Secondary school | 127 (38) |
| College/university | 94 (28) |
| Other/missing/don't know | 10 (3) |
| Median household size (IQR (range)) | 6 (4 (4–8)) |
| Total yearly household expenditure (US$) | $3569 |
| Number below national poverty line prior to illness, n (%) | 151 (45) |
| **Surgical care episode descriptors** | |
| Median days of sickness before presentation Median (IQR [range]) | 2 (14 [0–14]) |
| Number that sought care for illness elsewhere prior to presentation at Connaught, n (%) | 225 (67) |
| Mode of transport used to travel to hospital, n (%) | |
| Public transport | 241 (72) |
| Ambulance | 67 (20) |
| Private transport | 23 (7) |
| Walked | 3 (1) |
| Don't know/missing | 1 (0) |
| Emergency admission, n (%) | 242 (72) |
| Eligible for free healthcare, n (%)* | 70 (21) |
| Primary diagnosis by surgical condition, n (%) | |
| Trauma | 114 (34) |
| Hernia | 58 (17) |
| Abdominal conditions | 56 (17) |
| Peripheral vascular disease or diabetic foot disease | 27 (8) |
| Urological conditions | 23 (7) |
| Breast mass/cancer | 16 (5) |
| Burns | 15 (5) |
| ENT/dental disease | 13 (4) |
| Goitre | 7 (2) |
| Congenital abnormality (paediatrics) | 3 (1) |
| Missing/don't know | 3 (1) |
| Treatment, n (%) | |
| Operative | 226 (68) |

Continued

## Table 1 Continued

| | |
|---|---|
| Non-operative | 109 (33) |
| Median length of hospital stay in days (IQR (range)) | 8 (18 (3–21)) |

*Eligible for free healthcare indicates those that fall under the government Free Healthcare Initiative, a health financing policy introduced in 2010 aimed to significantly improve maternal and child health through the provision of free healthcare services for all children under 5 years, pregnant and lactating women. This was later extended to include Ebola survivors.
ENT, ear, nose and throat.

non-medical costs. Indirect costs, such as lost wages, totalled US$46 (table 2 and online supplemental appendix table 4).

Of the in-hospital direct medical costs, 48% were given to hospital staff (it was not clear whether the hospital staff later transferred these funds to the hospital bank or not), 33% were made directly to the hospital bank/cashiers and 17% to an external facility such as external pharmacy or diagnostic centre (online supplemental appendix table 5).

A variety of means were used to meet costs, and participants were allowed to mention more than one means of covering costs (table 3). Most (84% of patients) used their savings to meet some or all of the costs, with family contributions, borrowing money and charitable donations forming the second, third and fourth most frequently used means of meeting OOP payments, respectively. Only 2% (six patients) had some form of health insurance. Wider implications included loss of wages in 37% and loss of job in 6.0%.

CE, when defined as OOP costs of more than 10% of all household expenditure, affected 18% of those interviewed. In the sensitivity analysis using the threshold of more than 40% of non-subsistence expenditure, CE affected 10% of those interviewed.

Prior to the surgical care episode, 45% of people interviewed were below the national poverty line, 90% were below the World Bank poverty level and 70% below the World Bank extreme poverty level. Following payment for surgical care, 50% were pushed below or further below the national poverty line. Corresponding figures were 91% and 73%, for the World Bank thresholds of poverty and extreme poverty, respectively.

Analysis of the possession of household assets demonstrated that those interviewed were more likely to have electricity, a mobile phone, radio, television, refrigerator, bicycle, motorcycle or car than those of the general population in Sierra Leone (2015 census data, all p≤0.001) or of the urban population in the western area (2015 census data, all p≤0.05) (table 4).

Regression analysis demonstrated that the factors associated with greater costs were older age, longer length of hospital stay and undergoing a general surgical or urological procedure (online supplemental appendix table 6).

**Table 2** Out-of-pocket (OOP) costs

| Costs | Imputed mean cost (US$ (% of subtotal)) |
|---|---|
| **Prehospital costs** | |
| Direct prehospital medical OOP costs (total) | 21 (88 of 24) |
| ▶ Consultation | 2 (10 of 21) |
| ▶ Medications | 12 (57 of 21) |
| ▶ Medical supplies | 2 (10 of 21) |
| ▶ Investigations | 4 (19 of 21) |
| ▶ Other miscellaneous | 2 (10 of 21) |
| Direct (prehospital) non-medical OOP costs (total) | 3 (13 of 24) |
| ▶ Transport | 3 (100 of 3) |
| *Total prehospital costs* | *24 (10 of 243)* |
| **In-hospital costs** | |
| Direct medical OOP costs (total) | 138 (63 of 219) |
| ▶ Administrative | 20 (14 of 138) |
| ▶ Medications | 26 (19 of 138) |
| ▶ Medical supplies | 14 (10 of 138) |
| ▶ Investigations | 15 (11 of 138) |
| ▶ Blood transfusion | 9 (7 of 138) |
| ▶ Total operation costs | 49 (36 of 138) |
| ▶ Unofficial costs | 6 (4 of 138) |
| Other/miscellaneous | 1 (1 of 138) |
| Direct non-medical costs (total) | 34 (16 of 219) |
| ▶ Transport to hospital | 7 (21 of 34) |
| ▶ Food | 20 (59 of 34) |
| ▶ Accommodation | 0 (0 of 34) |
| ▶ Other* | 7 (21 of 34) |
| Indirect costs | |
| ▶ Lost wages | 46 (100 of 46) |
| Total OOP costs | 243 |

*Other relates to travel and other associated costs incurred as a result for needed investigations from and or medication/supplies from an external facility. SPSS calculates only the mean using imputed variables; hence, no SD is displayed.

## Discussion

In this study, we found that accessing and receiving tertiary-level surgical care in Sierra Leone requires large upfront OOP payments that have a substantial impact on individual and households' economic situations. These equate to a catastrophic expense in nearly a fifth of households and are impoverishing half of the households that receive care. We found poverty, as assessed by household expenditure, was high, indicating a limited financial buffer to accommodate costs of care. This is despite most people who access surgical care owning a higher level of assets than the general population.

**Table 3** How costs are met and the wider implications of seeking and undergoing surgical care (n is the number of cases with data on each variable)

| How costs were met (total number responding to question) | Number (%) that used this as a means of meet OOP costs |
|---|---|
| Used savings (n=326) | 273 (84) |
| Arranged family contributions (n=331) | 128 (39) |
| Borrowed money (n=331) | 102 (31) |
| Received charity money (n=331) | 83 (25) |
| Sold possessions (n=329) | 17 (5) |
| Other (n=331) | 14 (4) |
| Pawned possessions (n=332) | 8 (2) |
| Have health insurance (n=335) | 6 (2) |
| **Wider implications** | **Number (%) that experienced the wider implications of meeting OOP costs** |
| Loss of wages (n=328) | 121 (37) |
| Lost their job/changed their role at work/home (n=331) | 20 (6.0) |
| Disruption to education (n=333) | 12 (4) |

OOP, out-of-pocket.

The majority of the OOP payments were incurred in-hospital and as a result of direct medical costs. Payment for the operation itself and medications, medical supplies and investigations (including laboratory tests) were the biggest contribution to these costs. A small percentage of costs were categorised as unofficial, such as for 'nursing care' and 'tips', although these were given by a majority of people who received care. In addition, almost half of these were being paid through unofficial payment channels and made directly to staff. We do not know whether these payments were later transferred to the hospital

**Table 4** Ownership of household assets in comparison to 2015 census data

| Household assets | Surgical cohort Number (%) of households that own the asset | 2015 census data Whole country data (%) |
|---|---|---|
| Electricity | 227 (67.8) | 17.8 |
| Mobile phone | 326 (97.3) | 62.94 |
| Radio | 280 (83.6) | 58.03 |
| Television | 212 (63.3) | 19.76 |
| Refrigerator | 119 (35.5) | 8.22 |
| Bicycle | 38 (11.3) | 6.43 |
| Motorcycle | 8 (14.3) | 7.62 |
| Car | 50 (14.9) | 3.65 |

bank; however, these informal routes are common and indicate poor financial governance that urgently needs to be addressed.

The majority of payments were met using savings, followed by raising money from family or borrowing money. In addition, a large number of participants lost wages during the sickness episode and a proportion lost their jobs. In a country where informal work predominates and earnings can be unpredictable, this may impact on household financial security and influence future health seeking behaviour, both of the individuals affected and their immediate family and communities.

The majority of patients accessing surgical care were young males; whether this male predominance is a true reflection of surgical disease burden, beyond obstetrics and gynaecological care, in Sierra Leone or reveals a hidden gender bias in care-seeking behaviour is beyond the remit of this study. Nevertheless, males who sought care in our study are traditionally the main breadwinners and the most economically active population group in Sierra Leone. This loss of wages and livelihood could have implications on the wider socioeconomic determinants of health and the well-being of the household. The additional burden to the patients and their households as a result of the indirect costs supports the macroeconomic argument for investing in surgical care put forward by Grimes *et al*, who demonstrated the opportunity to avert 36 487 Disability Adjusted Life Years (DALYs) by investing in surgical care at hospital level in Sierra Leone.[23 24]

Some specialties, such as general surgery and urology, incurred much higher overall costs for the surgical episode, and this may be because operative intervention (with blood transfusion and a longer length of stay) is usually required. This contrasts, for example, with trauma care that was often managed non-operatively. Such non-operative treatment for trauma may be partly as a result of local surgical practice, often driven by lack of resources such as the unavailability of internal fixation wires and orthopaedic implants, and partly because some common orthopaedic problems are managed non-operatively. In addition, we found that age and length of hospital stay were associated with significantly higher costs. This may be due to the fact that those under the age of 5 years were eligible for free healthcare in Sierra Leone and that a longer stay in hospital was associated with higher direct non-medical and indirect costs such as payment for food and lost wages.

There are a limited number of studies to draw a direct comparison with as only a few used a similar methodology (direct interview) as opposed to modelled data or the use of caesarean section costs as a proxy measure to extrapolate costs, CE and impoverishment.[2 16 25–30] There are even fewer studies that report on the financial implications of all or most types of surgical care. The majority report on single surgical subspecialties such as obstetric care, paediatric surgery or trauma care. Nevertheless, there have been three recent studies from Uganda reporting CE rates of 31% and 55% and IE of 47%.[16 31 32] A study

in Malawi interviewing patients undergoing hernia operations reported CE rates as high as 90% using a threshold of 10% of yearly income.[28] Various studies looking at injury and trauma care costs in Vietnam, India and Nigeria have reported CE rates of 60%, 30% and 86%, respectively,[25] and a study in Morocco looking at obstetric surgical care alone estimated CE rates of 88%,[33] while an emergency obstetric care study in Indonesia estimated CE at 68%.[34] The intercountry variability makes it difficult to draw comparative conclusions. This highlights the need for a standardised way of assessing and measuring the financial implications of surgical care to allow accurate collection and reporting of these global surgery metrics on financial risk protection.

In keeping with other studies, we noted lower rates of CE and IE in comparison with the modelled and extrapolated estimates for Sierra Leone. This is probably because the modelled studies are based on the whole population that may require surgery and not on those that have successfully accessed surgical care. The lower rates of IE and CE seen may therefore be explained by a lack of access by the poorest. This is supported by data from Sierra Leone that estimates that up to 25% of deaths in 2011 could have been averted through access to safe, timely and affordable surgical care and that Sierra Leone has an unmet surgical burden of disease of 92%,[10] with approximately 70% of Sierra Leoneans stating that the financial burden of OOP payments for healthcare was the biggest barrier to accessing care.[35 36] In addition, we found that those accessing tertiary-level surgical care came from predominantly urban areas of Sierra Leone and, when compared with the wider Sierra Leone population, had significantly higher asset ownership. It may be therefore that the poorest and those at the highest risk of financial catastrophe are not accessing care when needed. This may also reflect other known barriers to seeking surgical care in LMICs that are often complex and multifactorial such as cultural beliefs, attitudes and fears towards surgical care and structural barriers such as geographical access, transport links and referral systems.[37]

### Limitations

There are several limitations to this study. First, it was dependant on recall and self-reported estimates of OOP costs and household expenditure. Although the questionnaire and methodology are a well-established way of obtaining this information in a low-resource setting where informal work predominates and payments are not often receipted. To increase accuracy of data collected, household expenditure questions were broken down to weekly, monthly and yearly costs, a chronological approach was used to the OOP cost questions that helped map out the patients' journey for them, participants were encouraged to bring an appropriate family member to the interview, in-country consensus gained and the questionnaire piloted prior to use.

Second, given that patients were often interviewed on the wards and potentially within hearing range of nurses,

data on informal payment methods and informal costs may not have been fully reported. If this were the case, we would have expected to see more missing data for payments made directly to staff in comparison with those made to the banks; however, we did not observe this. This suggests that participants did not appear to be deterred from sharing this information.

Third, the study only measured costs incurred during the illness episode up until discharge. We have therefore likely substantially underestimated the total costs of seeking surgical care.

In addition, in Sierra Leone tertiary-level obstetric care is provided at a different hospital and offered free of charge. Therefore, costs of accessing this were not included in this study. Further work needs to be done to see if those receiving free maternal healthcare incur any OOP costs and if informal payments such as tips paid to staff are as prevalent in the obstetric care hospital.

Finally, the desired sample size was not achieved as not all surgical patients admitted were interviewed. This was mostly due to many being discharged out of hours, at the weekend or after a short admission on the acute trauma ward before the study team could consent or interview them. This may indicate that these patient had minor pathology, a shorter stay and lower OOP costs. Inclusion of these cases may have lowered the mean OOP costs, CE and IE rates but would poorly represent the financial barriers and wider implications of accessing surgical care for those that may have absconded or self-discharged due the cost of care. Nevertheless, although sample size was not obtained, the 95% CI for a CE rate of 18% was 14% to 22%, which gives the study an overall power of 90%.

## Conclusion

This is the first empirical study from Sierra Leone that quantifies the financial burden of accessing and receiving surgical care. It adds insight into the global and national Sierra Leone modelled estimates of the likelihood of catastrophic and IE if surgery is required and joins the small but growing body of other empirical studies reporting on the OOP costs and wider financial implications of surgical care. In addition, it highlights the need to prioritise financial risk protection within healthcare and surgery if universal health coverage is to be achieved.

**Author affiliations**
[1] Department of Surgery, West Hertfordshire Hospitals NHS Trust, Watford, UK
[2] Faculty of Life Sciences and Medicine, King's College London, London, UK
[3] Department of Surgery, Medway NHS Foundation Trust, Gillingham, UK
[4] Department of Surgery, University of Sierra Leone College of Medicine and Allied Health Sciences, Freetown, Sierra Leone
[5] College of Medicine and Allied Health Sciences, University of Sierra Leone, Freetown, Sierra Leone
[6] King's Centre for Global Health, King's College London Faculty of Life Sciences and Medicine, London, UK
[7] Centre of Global Surgery, Department of Global Health, Stellenbosch University, Cape Town, South Africa
[8] Institute of Applied Health Research, University of Birmingham, Birmingham, UK

**Acknowledgements** We would like to thank the healthcare workers and patients who were involved in refining the data collection tool to ensure its applicability to a local setting.

**Contributors** JD, AJML, TBK and HW conceptualised the study. MP, JD and AJML developed the protocol and survey tools; MP, JD and CEG analysed the data; all authors contributed to the interpretation of the results and write up of the manuscript; all authors approved the manuscript for publication.

**Funding** This research was partly funded by the National Institute of Health Research (NIHR) Global Health Research Unit on Health System Strengthening in Sub-Saharan Africa, King's College London (GHRU 16/136/54) using UK aid from the UK Government to support global health research.

**Disclaimer** The views expressed in this publication are those of the author(s) and not necessarily those of the NIHR or the Department of Health and Social Care.

**Competing interests** None declared.

**Patient consent for publication** Not required.

**Ethics approval** Ethical approval was granted by the Sierra Leone Ethics and Scientific Review Committee and from the King's College London Research Ethics Committee (ref. LRU-17/18–6455).

**Provenance and peer review** Not commissioned; externally peer reviewed.

**Data availability statement** Data are available on reasonable request. Further data are available on reasonable request from the corresponding author.

**ORCID iD**
Caris E Grimes http://orcid.org/0000-0002-1662-5799

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
