## [Reviewer comments · BMJ Open]

ARTICLE DETAILS

TITLE (PROVISIONAL)	What is the financial burden to patients of accessing surgical care in Sierra Leone? A cross-sectional survey of catastrophic and impoverishing expenditure
AUTHORS	Phull, Manraj; Grimes, Caris; Kamara, Thaim; Wurie, Haja; Leather, Andy; Davies, Justine

VERSION 1 – REVIEW

REVIEWER	Leon Bijlmakers Radboud UMC Nijmegen, The Netherlands
REVIEW RETURNED	23-Apr-2020

GENERAL COMMENTS	A well written paper, well structured, well argued, supported by sound data, well referenced. A few comments, questions, suggestions for your consideration: • Presentation of mean versus median (e.g. median hh expenditure and mean OOP): the explanation on p7 (lines 35-37) is well taken – means + SD are presented for normally distributed data; otherwise medians and IQR. In the latter case, however, it would be better to present the range as well (throughout the manuscript, including the tables where appropriate).• Results, p9: “83.7% of the patients used their savings to meet some or all of the costs.” This is a bit misleading. From Table 3 I understand that interviewees were allowed (and rightly so) to mention more than one manner in which they covered the costs. It would therefore be appropriate to say that in the text; and/or to mention the 2nd, 3rd and 4th most frequently mentioned sources (i.e. family contributions, borrowed money from someone, received a donation/charity) – as you have done in the Discussion section (p10, line 25).• Discussion section, p10, lines 53-54: “... this may be due to the fact ... that a longer stay in hospital may be associated with higher direct non-medical and indirect costs ...”. Apparently (as mentioned on p7, lines 15-16) you did record/calculate length of stay since admission to the hospital for all interviewed patients. Therefore I would assume you can verify whether there is any such association.• Discussion section, “Analysis of the missing data”, p12, lines 3-4: does that refer to the Multiple Imputation Chained Equations method that you describe in the Methods section (p8), which I’m personally not familiar with? Is there any table or other data set that you could bring in to substantiate the statement that this
---

	analysis "... did not show a correlation or bias towards more missing costs if paid if paid directly to staff versus at the bank"?  • Study limitations (in Discussion section and Abstract): - A bit more reflection is required on the extent to which people in need of surgical care forego such care because of expense – and the relative weight of barriers that prevent people from seeking care (fin vs geographic vs cultural). - (re Fig 1): a large number of surgical patients were not interviewed (230 + 81), mainly because they were discharged before their consent could be obtained, or they could not be located on the study wards after their admission; or they were lost from the study after consenting (81). Could they be any different from the ones interviewed? (E.g. less serious cases, shorter duration of stay.) And if so, how could this have influenced the results? - The fact that (according to the results presented on p9) the main breadwinner of 28% and 38% of the interviewed patients had college/university education and secondary education, respectively, and that 29% of interviewees were in formal employment makes me believe that this sample was a bit of an extraordinary group. Could you validate that by comparing with data from other national surveys (similar to what Table 4 is showing)? - You actually do reflect (on p11, last paragraph before Limitations) on differential access to surgery – which is good – but you could bring in the results from earlier studies about the scope and extent of unmet surgical need (untreated surgical) conditions, such as the study by Groen et al (ref #10). Minor comment:  • Introduction, p 5, line 36: "... payments for health care are upfront ...". It would be good to explain, e.g. in a (sub-)sentence, whether or not patients pay an admission fee, and an additional amount per night stayed at the hospital. Wish you success!
--	--

REVIEWER	Charles Liu, MD, MS Department of Surgery, Stanford University Stanford, California, United States
REVIEW RETURNED	07-May-2020

GENERAL COMMENTS	In this prospective survey study of 335 patients undergoing surgery at the main tertiary referral hospital in Freetown, Sierra Leone between June and August of 2018, the authors find that 18% of patients incurred catastrophic expenditures and 11% of those not already in poverty incurred impoverishing expenditures due to the costs of surgery. They should be applauded for this thorough and well-planned effort to collect real-world data on out-of-pocket spending for surgical care in a resource-deprived setting. Their work is particularly important as the Lancet Commission on Global Surgery has identified catastrophic and impoverishing expenditures as 2 of its 6 key surgical indicators, but real-world data on these outcomes are rarely available in LMICs. I believe this article is a valuable addition to the literature and hopefully can also be fed back to Sierra Leonean stakeholders to guide policy. I have some comments and suggestions:
---

	1. The article is currently about 4400 words long. For readability, I would recommend that the authors shorten the manuscript by about 1000 words, if possible. For example, instead of a paragraph-form description of data elements collected from study participants, it would be better to briefly summarize the data elements, then include the detailed data collection form in the appendix. This would also improve reproducibility of this study in other settings. Words may also be cut by shortening sentences. 2. Also for readability, would recommend avoiding use of acronyms when not necessary. For example, the acronym “WB” is used but not defined. Furthermore, SL seems like an unnecessary abbreviation for Sierra Leone. CE, IE, OOP, and other specific terms defined clearly by the authors are reasonable to include. 3. I would recommend that the authors adopt one of the two definitions of CE as primary, and designate the other as a secondary or sensitivity analysis (although both are widely used and important to include). This would improve interpretability and comparability to prior studies, such as the CE study from Uganda (Anderson et al. 2017). Furthermore, as currently written, I initially thought the authors defined CE as patients meeting either the 10% or the 40% spending threshold, while in fact they applied these two thresholds separately. Designating one as primary would clarify this. 4. The authors report that 45% of patients were already below the national poverty line and 50% were pushed into or further into poverty by surgical expenses. This strikes me as a counterintuitive way to present the impoverishing effect of surgery, as even a \$0.10 expense would then be “impoverishing” for someone already in poverty. An alternative may be to state that 45% of patients were already below poverty, and 11% of those who were not (5%/55%) were pushed below the poverty line due to surgery. 5. In the sentence beginning “Regression analysis demonstrated that the factors that were significantly associated with increased costs were...”, please clarify the direction of the association, for example “factors that were associated with greater costs were higher age, longer length of hospital stay...” 6. I appreciate the discussion of why this study found lower rates of CE than modeled studies of surgical patients in Sierra Leone. However, do the authors have any hypotheses on why they found lower rates of CE (18%) than comparable studies in Uganda (31%) and elsewhere? Do they feel this lower rate is real and reflects better financial protection conferred by the Sierra Leone healthcare system? Or conversely, that even fewer of the poorest patients are reaching and using surgical care in Sierra Leone? Or differences in measurement?
--	---

REVIEWER	Carlos Shiraishi-Zapata Servicio de Centro Quirúrgico y Anestesiología Hospital EsSalud Talara, Piura-Perú
REVIEW RETURNED	16-May-2020

GENERAL COMMENTS	Dear members of the BMJ Open Editorial Office,
--

	First of all, I would like to thank you for the honour of reviewing this manuscript. The suggestions and comments for the manuscript's authors are the following: Methods Setting It is suggested to write the meaning of the abbreviation ENT. Data collection What was the reason for choosing to carry out a pilot study of the characteristics of the questionnaire in 6 patients? Please, explain about the reasons. Definition and construction of variables The third paragraph presents important information. However, it is extensive and difficult to follow. I suggest that it be divided or supported by a diagram that makes the information easier to understand. In the formulas of Baseline poverty (BLPh) and Impoverishment expenditure, it would be advisable to write the meaning of the abbreviation. Sample size and power calculation Does the abbreviation WB mean "World Bank"? Results Table 1: Participant characteristics What is the difference between "Eligible for free health care (%) - 20.9%" and a patient with insurance? Did this category (Eligible for free health care) include patients listed as insured-6%? Please, address this point with more information. Appendix table 5 Please, put in the table legend the meaning of the abbreviation "GLM."
--	--

REVIEWER	Rolvix Patterson Duke University Hospital, United States of America
REVIEW RETURNED	23-May-2020

GENERAL COMMENTS	Peer Review: Catastrophic and impoverishing expenditure for surgical care in Sierra Leone Surgery has been recognized as a component of Universal Health Coverage; however, many barriers exist to achieving broad access to surgical care. It has been noted that affordability "is one of the major barriers of access to surgery."¹ Despite the inclusion of catastrophic and impoverishing expenditure within the World Development Indicators, there has been a paucity of primary data on OOP expenditures that has compelled surgical policy and interventions to rely on modeled estimates. This study seeks to address this gap in Sierra Leone by using a cross-sectional survey and statistical analysis to determine amount of out-of-pocket (OOP) costs and rates of catastrophic expenditure, impoverishment, and means to meet these costs. The authors find that patients who seek surgical care at Connaught Hospital face
--

substantial direct and indirect expenses which frequently push them into poverty. They conclude that financial risk protection for surgery must be prioritized in surgical scale-up and efforts made to achieve Universal Health Coverage. My comments are as follows:

Title and Abstract

These are appropriate and accurate.

Introduction

The introduction adequately frames the global burden of catastrophic expenditure for surgery and the general need for financial risk protection. The authors effectively argue the need for additional primary data on OOP costs. They appropriately frame this need within the costing literature specific to Sierra Leone and the policy window presented by the National Surgical, Obstetric, and Anesthesia Plan (NSOAP). They provide adequate justification of their methodology. Furthermore, the authors clearly outline their research questions.

Financial risk protection (FRP) is defined twice in the first paragraph – only the first one is necessary.

Methods

The study setting is well-described. Participant inclusion and exclusion criteria were clear and comprehensive. The authors describe the development of the study questionnaire and highlight the essential step of co-designing this with in-country experts. The collected variables are appropriate and satisfy previous recommendations, e.g. “[future research] should incorporate all costs borne out-of-pocket by patients, whether they are fees for the procedure itself, costs for the procurement of the necessary drugs and supplies, or the costs of transportation, food and lodging necessary to obtain care.”² The definitions of OOP and CE are clear. Poverty thresholds are outlined, and the findings are strengthened by the inclusion of the Sierra Leone national poverty line.

There is no description of how the training of Sierra Leonean research assistants were trained – this would help the reader better evaluate the reliability of the results. I also recommend including the questionnaire in the appendix to support similar research endeavors in other locations and allow for critical evaluation by the readers.

The sample size, power calculation, and statistical analysis appear sound. However, I do not have expertise in Multiple Imputation Chained Equations, so a statistician should be consulted to evaluate this analysis.

This study appears to have been conducted ethically. Importantly, approval was granted by both King's College and the Sierra Leone Ethics and Scientific Review Committee. Provisions were made for patients who were illiterate. Informed consent was conducted thoughtfully and appropriately.

Results

The authors present the results clearly and succinctly.

Discussion and Conclusion

The discussion highlights and contextualizes the key results, raises relevant questions, and situates the findings within the broader literature. The limitations are thorough and well-presented.

Given the conclusion that financial risk protection is needed, it would could strengthen the article to include a discussion on avenues to improve FRP (e.g. insurance) and financing models to reduce OOP costs (e.g. cash transfers, fee stratification, etc).

Citations

Citations 11 and 20 provide no website/journals or access dates.

Tables, Figures, and Appendices

Figure 1, Table 1, Table 3, Table 4, and Appendix 1 are clear and convey the relevant data.

The parenthesis in Table 2 and Appendix table 3 are difficult to interpret at first glance. These could be made clearer by removing the parentheses entirely for subtotals, dropping the denominator in parenthesis for individual costs (e.g. 21 for direct pre-hospital medical OOP cost), and clarifying with a reference example (in the heading rows) like: \$US (% of subtotal).

Appendix 2 reports data from the 2014 Sierra Leone Economic and Financial Survey. I don't see this cited anywhere.

	Furthermore, though you specify in the text that costs are “presented in Le and \$US at the conversion rate of 15th July 2019,” it is not evident that the costs from the Economic and Financial Survey were similarly adjusted for inflation despite the direct comparison that you make. On review of the referenced survey, it appears that these values are as-reported in 2014. Your assertion that your study population has a significantly higher proxy of wealth or socioeconomic status rests on your analysis of assets, and I do not think that this should be affected. However, without either 1) adjusting the Economic and Financial Survey results for inflation, or 2) adding a disclaimer that these data cannot be compared directly, I think that Appendix 2 might leave the reader with the understanding that wealth/socioeconomic status is inordinately higher in your study population than the broader national population. This may be an artifact of the submission process, but the appendices are listed in duplicate (p24-28). Summary This is a well-written article that contributes to a key gap in the health systems and global surgery literature. I am impressed by this work, and my comments and suggested revisions are all minor. As such, I recommend this article for publication with minor revisions.
--	--

REVIEWER	Lina Roa Department of Obstetrics & Gynecology, University of Alberta, Edmonton, Canada
REVIEW RETURNED	23-May-2020

GENERAL COMMENTS	Thank you for the opportunity to review this manuscript. This manuscript answered an important question on the financial burden associated with surgical care in Sierra Leone using rigorous methodology and addresses the need to move beyond modelled FRP data. Overall, it is a robust manuscript. My main concern is in the exclusion of obstetrics and gynecology surgery, particularly around caesarean sections, since they are one of the most commonly performed procedures, represents a large proportion of the surgical volume in LMICs and often are provided free of charge. The omission of obstetric care requires more explanation and it potentially needs to be recognized as a limitation. A few more detailed points are included below. Abstract -Methods: CE is defined as >40% of capacity to pay. Elsewhere in the paper the definition used is “>40% non-subsistence expenditure” which in my opinion is clearer. Consider changing the definition in the abstract. Intro -Line 23/24 needs clarification. Were the modelling studies in SL? Or in all of West Africa? Or are the authors just saying that SL is in West Africa?
---

	-Last paragraph. The metrics were nicely described in a-d. It is not clear if the in-hospital payment mechanism, how costs were met and factors associated were also obtained from the exit survey and if they were also aims of the study or not. Please clarify Methods -Setting: Please explain why obstetrics & gynecology were excluded given that caesarean sections are one of the most commonly performed surgical procedures. Were these not provided in this hospital? - Data collection: Please clarify if the questionnaire was adapted from existing ones (ref 16-19) or if a whole new questionnaire was developed. If the later, please explain what was better about the new questionnaire developed. Are any of the existing questionnaires validated? -Detailed definition of variable is great! -Is there a reason why for indirect costs only lost wages were considered? Were other costs such as those to caregivers considered? -For lost wages what time period was included? Only hospitalization? Or was the recovery time included too? - Page 7: How was the conversion from LE to USD done? Was PPP used? Results -Table 2: Please clarify what is being presented in the brackets after the imputed mean cost Discussion: Overall very well written. -Page 10, line 32-35: If obstetrics and gynecology surgery had been included it is likely that the majority of patients accessing surgical care might not have been males. -Page 11, line 13-14. The Ugandan study referred to include cesarean sections and found that patients that underwent cesareans were less likely to experience a catastrophic expenditure compared to those who didn't. Is it possible that if you had included cesarean sections in your study, that the risk of CCE and IE would have been less?
--	---

VERSION 1 – AUTHOR RESPONSE

Reviewer(s)' Comments to Author:

Reviewer: 1

Reviewer Name: Leon Bijlmakers

Institution and Country: Radboud UMC Nijmegen, The Netherlands

Please state any competing interests or state 'None declared': None declared

Please leave your comments for the authors below

A well written paper, well structured, well argued, supported by sound data, well referenced.

Thank you!

A few comments, questions, suggestions for your consideration:

- Presentation of mean versus median (e.g. median hh expenditure and mean OOP): the explanation

on p7 (lines 35-37) is well taken – means + SD are presented for normally distributed data; otherwise medians and IQR. In the latter case, however, it would be better to present the range as well (throughout the manuscript, including the tables where appropriate).

We have added IQR's and ranges to all the medians.

- Results, p9: “83.7% of the patients used their savings to meet some or all of the costs.” This is a bit misleading. From Table 3 I understand that interviewees were allowed (and rightly so) to mention more than one manner in which they covered the costs. It would therefore be appropriate to say that in the text; and/or to mention the 2nd, 3rd and 4th most frequently mentioned sources (i.e. family contributions, borrowed money from someone, received a donation/charity) – as you have done in the Discussion section (p10, line 25).

This has been done. See: “A variety of means were used to meet costs and participants were allowed to mention more than one means of covering costs (Table 3). Most (83.7% of patients) used their savings to meet some or all of the costs, with family contributions, borrowing money and charitable donations forming the 2nd, 3rd and 4th most frequently used means of meeting OOP payments, respectively. Only 2% (6 patients) had some form of health insurance. Wider implications included loss of wages in 36.9% and loss of job in 6.0%.”

- Discussion section, p10, lines 53-54: “... this may be due to the fact ... that a longer stay in hospital may be associated with higher direct non-medical and indirect costs ...”. Apparently (as mentioned on p7, lines 15-16) you did record/calculate length of stay since admission to the hospital for all interviewed patients. Therefore I would assume you can verify whether there is any such association.

We have changed the wording on this as our regression analysis did show an association. See: “This may be due to the fact that those under the age of 5 years were eligible for free health care in Sierra Leone and that a longer stay in hospital was associated with higher direct non-medical and indirect costs such as payment for food and lost wages.”

- Discussion section, “Analysis of the missing data”, p12, lines 3-4: does that refer to the Multiple Imputation Chained Equations method that you describe in the Methods section (p8), which I'm personally not familiar with? Is there any table or other data set that you could bring in to substantiate the statement that this analysis “... did not show a correlation or bias towards more missing costs if paid if paid directly to staff versus at the bank”?

We have changed the wording on this so as to make it clearer, as this doesn't refer to the statistical handling of missing data. See: “Secondly, given that patients were often interviewed on the wards and potentially within hearing range of nurses, data on informal payment methods and informal costs, may not have been fully reported. With this in mind we would have expected to see more missing data for the variable payments made directly to staff in comparison to those made to the banks, however we did not observe this. This indicates that participants were not deterred from sharing information on informal payments within the in-hospital study setting.”

- Study limitations (in Discussion section and Abstract):
 - A bit more reflection is required on the extent to which people in need of surgical care forego such care because of expense – and the relative weight of barriers that prevent people from seeking care (fin vs geographic vs cultural).

We have added further reflection to the last paragraph in the discussion to address this. See “This may also reflect other known barriers to seeking surgical care in LMICs that are often complex and

multifactorial such as cultural beliefs, attitudes and fears towards surgical care and structural barriers such as geographical access, transport links and referral systems.”

- (re Fig 1): a large number of surgical patients were not interviewed (230 + 81), mainly because they were discharged before their consent could be obtained, or they could not be located on the study wards after their admission; or they were lost from the study after consenting (81). Could they be any different from the ones interviewed? (E.g. less serious cases, shorter duration of stay.) And if so, how could this have influenced the results?

Additional comments have been added to the discussion limitations to address this. See: “Finally, the desired sample size was not achieved as not all surgical patients admitted were interviewed. This was mostly due to many being discharged out of hours, at the weekend or after a short admission on the acute trauma ward, before the study team could consent or interview them. With regards to the later this may indicate minor pathology, a shorter stay and therefore lower OOP costs. Inclusion of these cases may have lowered the mean OOP costs, CE and IE rates but would poorly represent the financial barriers and wider implications of accessing surgical care for those that may have absconded or self-discharged due the cost of care..”

- The fact that (according to the results presented on p9) the main breadwinner of 28% and 38% of the interviewed patients had college/university education and secondary education, respectively, and that 29% of interviewees were in formal employment makes me believe that this sample was a bit of an extraordinary group. Could you validate that by comparing with data from other national surveys (similar to what Table 4 is showing)?

The 2015 Census data only asks about the population currently in primary level education and is not comparable. We have already shown that our population is wealthier than the average within the country – but we have done this with asset data rather than education (please see table 4).

- You actually do reflect (on p11, last paragraph before Limitations) on differential access to surgery – which is good – but you could bring in the results from earlier studies about the scope and extent of unmet surgical need (untreated surgical) conditions, such as the study by Groen et al (ref #10).

See response to previous comment about barriers to access.

Minor comment:

• Introduction, p 5, line 36: “... payments for health care are upfront ...”. It would be good to explain, e.g. in a (sub-)sentence, whether or not patients pay an admission fee, and an additional amount per night stayed at the hospital.

This introductory paragraph is aimed to highlight complexities of OOP payments for health care in Sierra Leone as well as other LMICs and therefore has been left as a slightly broader statement. The upfront costs that are specific to SL are further characterised in the rest of the paper.

Wish you success!

Thank you!

Reviewer: 2

Reviewer Name: Charles Liu, MD, MS

Institution and Country:

Department of Surgery, Stanford University
Stanford, California, United States

Please state any competing interests or state 'None declared': None declared

Please leave your comments for the authors below

In this prospective survey study of 335 patients undergoing surgery at the main tertiary referral hospital in Freetown, Sierra Leone between June and August of 2018, the authors find that 18% of patients incurred catastrophic expenditures and 11% of those not already in poverty incurred impoverishing expenditures due to the costs of surgery. They should be applauded for this thorough and well-planned effort to collect real-world data on out-of-pocket spending for surgical care in a resource-deprived setting. Their work is particularly important as the Lancet Commission on Global Surgery has identified catastrophic and impoverishing expenditures as 2 of its 6 key surgical indicators, but real-world data on these outcomes are rarely available in LMICs. I believe this article is a valuable addition to the literature and hopefully can also be fed back to Sierra Leonean stakeholders to guide policy.

Thank you for this comment. We have shared these results with the Ministry of Health and Sanitation in Sierra Leone and are in active engagement with them about how to effect change.

I have some comments and suggestions:

1. The article is currently about 4400 words long. For readability, I would recommend that the authors shorten the manuscript by about 1000 words, if possible. For example, instead of a paragraph-form description of data elements collected from study participants, it would be better to briefly summarize the data elements, then include the detailed data collection form in the appendix. This would also improve reproducibility of this study in other settings. Words may also be cut by shortening sentences.

Many of the comments from the five reviewers are asking for additional information / clarification resulting in a longer rather than a shorter manuscript. However, we have cut down the section on definition and construction of variables and inserted the questionnaire as an additional appendix table. See "Data was collected on the participants age, gender and address (later used to calculate if they were resident in an urban or rural area). The occupation of the main breadwinner was recorded using free text followed by a question on whether this was salaried (i.e. employed) or non-salaried (i.e. self-employed or working in the informal sector). Education was captured as the highest level of education of the main breadwinner. Information on household expenditure was captured by asking 7 questions on regular items purchased in a typical week (food and drink etc.), 11 questions on larger expenditure items typically purchased monthly (toiletries, clothing, etc.) and a further 12 questions on typical yearly spend on big household items such as furniture and livestock (see Appendix 6)."

2. Also for readability, would recommend avoiding use of acronyms when not necessary. For example, the acronym "WB" is used but not defined. Furthermore, SL seems like an unnecessary abbreviation for Sierra Leone. CE, IE, OOP, and other specific terms defined clearly by the authors are reasonable to include.

This has been done.

3. I would recommend that the authors adopt one of the two definitions of CE as primary, and designate the other as a secondary or sensitivity analysis (although both are widely used and important to include). This would improve interpretability and comparability to prior studies, such as

the CE study from Uganda (Anderson et al. 2017). Furthermore, as currently written, I initially thought the authors defined CE as patients meeting either the 10% or the 40% spending threshold, while in fact they applied these two thresholds separately. Designating one as primary would clarify this.

Thank you - we have changed this so that it is clear that both thresholds were used in the estimates and are presented in the results.

4. The authors report that 45% of patients were already below the national poverty line and 50% were pushed into or further into poverty by surgical expenses. This strikes me as a counterintuitive way to present the impoverishing effect of surgery, as even a \$0.10 expense would then be “impoverishing” for someone already in poverty. An alternative may be to state that 45% of patients were already below poverty, and 11% of those who were not (5%/55%) were pushed below the poverty line due to surgery.

This has been done. See: “45% of patients were already below the national poverty line prior to admission, and 11% of those who were not were pushed below the poverty line following payment for surgical care.”

5. In the sentence beginning “Regression analysis demonstrated that the factors that were significantly associated with increased costs were...”, please clarify the direction of the association, for example “factors that were associated with greater costs were higher age, longer length of hospital stay...”

This has been done. See: “Regression analysis demonstrated that the factors associated with greater costs were older age, longer length of hospital stay and undergoing a general surgical or urological procedure (Appendix table 5).”

6. I appreciate the discussion of why this study found lower rates of CE than modeled studies of surgical patients in Sierra Leone. However, do the authors have any hypotheses on why they found lower rates of CE (18%) than comparable studies in Uganda (31%) and elsewhere? Do they feel this lower rate is real and reflects better financial protection conferred by the Sierra Leone healthcare system? Or conversely, that even fewer of the poorest patients are reaching and using surgical care in Sierra Leone? Or differences in measurement?

We feel as mentioned in the last 2 paragraphs of the discussion section that this is due to multiple reasons mostly centred around the fact that comparison is difficult due to the use of different methodology and a lack of standardised tools to measure CE / IE. In addition, the Ugandan study was performed in a government hospital where healthcare is meant to be free of charge, potentially resulting in improved access due to the perceived reduction in the financial barriers associated with care. However, as Anderson et al. demonstrated though intended to be free, OOP costs were incurred leading to IE and CE in a poor population who if in Sierra Leone may be deterred from seeking surgical care in the first instance.

Reviewer: 3

Reviewer Name: Carlos Shiraishi-Zapata

Institution and Country:

Servicio de Centro Quirúrgico y Anestesiología

Hospital EsSalud Talara, Piura-Perú

Please state any competing interests or state 'None declared': None declared

Please leave your comments for the authors below

Dear authors,

First of all, I would like to congratulate you for the effort made to carry out this study. Likewise, I have sent a small list of comments and suggestions that must be answered according to Editorial Office's instructions. See List of observations bmjopen-2020-039049.pdf:

Dear members of the BMJ Open Editorial Office,

First of all, I would like to thank you for the honour of reviewing this manuscript. The suggestions and comments for the manuscript's authors are the following:

Methods

Setting

It is suggested to write the meaning of the abbreviation ENT.

This has been done.

Data collection

What was the reason for choosing to carry out a pilot study of the characteristics of the questionnaire in 6 patients? Please, explain about the reasons.

A pilot of the questionnaire was performed by the research team with no set predetermined number of interviews. The piloting of the interview was an iterative process and the research team with trained research assistants felt that by the 6th interview they had gathered sufficient information to highlight any issues and changes that needed to be made as repeating patterns were identified. Based on this the pilot study was stopped after 6 patients had been interviewed.

Definition and construction of variables

The third paragraph presents important information. However, it is extensive and difficult to follow. I suggest that it be divided or supported by a diagram that makes the information easier to understand. In the formulas of Baseline poverty (BLPh) and Impoverishment expenditure, it would be advisable to write the meaning of the abbreviation.

All abbreviations have been defined in the text and including them in the formulas as well will make the formulas very large and difficult to represent clearly.

Sample size and power calculation

Does the abbreviation WB mean "World Bank"?

This has been changed.

Results

Table 1: Participant characteristics

What is the difference between "Eligible for free health care (%) - 20.9%" and a patient with insurance?

Did this category (Eligible for free health care) include patients listed as insured-6%? Please, address this point with more information.

Those eligible for free health care fall under a health financing government initiative introduced in 2010; the Free Health Care Initiative (FHCI). This was introduced to ensure a significant improvement in maternal and child health through the provision of free healthcare services for all children under 5, pregnant and lactating women and was later extended to include Ebola survivors. This does therefore not include those with insurance. This has been clarified in Table 1. See: "** Eligible for free health care indicates those that fall under the government Free Health Care Initiative (FHCI); a health

financing policy introduced in 2010 aimed to improve maternal and child health through the provision of free healthcare services for all children under 5, pregnant and lactating women. This was later extended to include Ebola survivors.”

Appendix table 5

Please, put in the table legend the meaning of the abbreviation “GLM.”

This has been done.

Reviewer: 4

Reviewer Name: Rolvix Patterson

Institution and Country: Duke University Hospital, United States of America

Please state any competing interests or state 'None declared': None declared

Please leave your comments for the authors below

Please see the attached file for comments. Peer Review BMJ Open.pdf:

Peer Review: Catastrophic and impoverishing expenditure for surgical care in Sierra Leone
Surgery has been recognized as a component of Universal Health Coverage; however, many barriers exist to achieving broad access to surgical care. It has been noted that affordability “is one of the major barriers of access to surgery.”¹ Despite the inclusion of catastrophic and impoverishing expenditure within the World Development Indicators, there has been a paucity of primary data on OOP expenditures that has compelled surgical policy and interventions to rely on modeled estimates. This study seeks to address this gap in Sierra Leone by using a cross-sectional survey and statistical analysis to determine amount of out-of-pocket (OOP) costs and rates of catastrophic expenditure, impoverishment, and means to meet these costs. The authors find that patients who seek surgical care at Connaught Hospital face substantial direct and indirect expenses which frequently push them into poverty. They conclude that financial risk protection for surgery must be prioritized in surgical scale-up and efforts made to achieve Universal Health Coverage. My comments are as follows:

Title and Abstract

These are appropriate and accurate.

Introduction

The introduction adequately frames the global burden of catastrophic expenditure for surgery and the general need for financial risk protection. The authors effectively argue the need for additional primary data on OOP costs. They appropriately frame this need within the costing literature specific to Sierra Leone and the policy window presented by the National Surgical, Obstetric, and Anesthesia Plan (NSOAP). They provide adequate justification of their methodology. Furthermore, the authors clearly outline their research questions.

Financial risk protection (FRP) is defined twice in the first paragraph – only the first one is necessary. This has been done.

Methods

The study setting is well-described. Participant inclusion and exclusion criteria were clear and comprehensive. The authors describe the development of the study questionnaire and highlight the essential step of co-designing this with in-country experts. The collected variables are appropriate and satisfy previous recommendations, e.g. “[future research] should incorporate all costs borne out-of-pocket by patients, whether they are fees for the procedure itself, costs for the procurement of the necessary drugs and supplies, or the costs of transportation, food and lodging necessary to obtain care.”² The definitions of OOP and CE are clear. Poverty thresholds are outlined, and the findings are strengthened by the inclusion of the Sierra Leone national poverty line.

There is no description of how the training of Sierra Leonean research assistants were trained – this would help the reader better evaluate the reliability of the results.

This has been added to the Appendix. See “Additional information on recruitment and training of research assistants

RAs were recruited through a competitive process and trained to administer the questionnaire. Training for all RAs was standardised and formally ran over 2 days. This involved; a formal presentation introducing the study, a review of all study processes and associated documents, a role play interview between the RAs using the questionnaire, a walk through the hospital to ensure the RAs gained an insight in to the surgical patients’ journey and points at which OOP payments may be made or cost incurred and a review of clinical notes, ward admission books and theatre log books to ensure that all demographic and diagnostic information was accurately captured.”

I also recommend including the questionnaire in the appendix to support similar research endeavors in other locations and allow for critical evaluation by the readers.

We have included the questionnaire to the Appendix. See Appendix table 6.

The sample size, power calculation, and statistical analysis appear sound. However, I do not have expertise in Multiple Imputation Chained Equations, so a statistician should be consulted to evaluate this analysis.

This study appears to have been conducted ethically. Importantly, approval was granted by both King’s College and the Sierra Leone Ethics and Scientific Review Committee. Provisions were made for patients who were illiterate. Informed consent was conducted thoughtfully and appropriately.

Results

The authors present the results clearly and succinctly.

Discussion and Conclusion

The discussion highlights and contextualizes the key results, raises relevant questions, and situates the findings within the broader literature. The limitations are thorough and well- presented.

Given the conclusion that financial risk protection is needed, it would could strengthen the article to include a discussion on avenues to improve FRP (e.g. insurance) and financing models to reduce OOP costs (e.g. cash transfers, fee stratification, etc).

Although interesting, we do not feel it is in the remit of this paper to embark on a complex discussion about avenues to improve financial risk protection – this would need to be a separate paper in its own right.

Citations

Citations 11 and 20 provide no website/journals or access dates.

Citation 11 has been removed along with the associated sentence and the following sentence changed to “To enable effective planning of surgical services in future, an accurate understanding of the financial implications of accessing surgical services is required.”

Citation 20 has been updated.

Tables, Figures, and Appendices

Figure 1, Table 1, Table 3, Table 4, and Appendix 1 are clear and convey the relevant data.

The parenthesis in Table 2 and Appendix table 3 are difficult to interpret at first glance. These could be made clearer by removing the parentheses entirely for subtotals, dropping the denominator in parenthesis for individual costs (e.g. 21 for direct pre-hospital medical OOP cost), and clarifying with a reference example (in the heading rows) like: \$US (% of subtotal).

We have clarified this with a reference example in the heading rows; \$US (% of subtotal).

Appendix 2 reports data from the 2014 Sierra Leone Economic and Financial Survey. I don’t see this cited anywhere.

This has been done (Reference 21).

Furthermore, though you specify in the text that costs are “presented in Le and \$US at the conversion rate of 15th July 2019,” it is not evident that the costs from the Economic and Financial Survey were similarly adjusted for inflation despite the direct comparison that you make.

Costs from the Economic and Financial Survey were not adjusted for inflation given that the categorisation of these data is not directly comparable, we felt that adjusting for inflation would give a spurious impression of comparability. We have included a disclaimer to that effect. See: “Appendix table 2: Household expenditure data. Variables on household expenditure shown here, for broad

comparison, with the Economic and Financial survey Sierra Leone 2014 data. Categories were harmonised where possible, however given differences in questions asked between surveys, an exact match of categories was not possible to achieve. Costs from the 2014 Economic and Financial Survey were not adjusted for inflation which needs to be considered when reviewing this data.”

On review of the referenced survey, it appears that these values are as-reported in 2014. Your assertion that your study population has a significantly higher proxy of wealth or socioeconomic status rests on your analysis of assets, and I do not think that this should be affected. However, without either 1) adjusting the Economic and Financial Survey results for inflation, or 2) adding a disclaimer that these data cannot be compared directly, I think that Appendix 2 might leave the reader with the understanding that wealth/socioeconomic status is inordinately higher in your study population than the broader national population.

We have included a disclaimer to Appendix 2 to that effect. See: “Appendix table 2: Household expenditure data. Variables on household expenditure shown here, for broad comparison, with the Economic and Financial survey Sierra Leone 2014 data. Categories were harmonised where possible, however given differences in questions asked between surveys, an exact match of categories was not possible to achieve. Costs from the 2014 Economic and Financial Survey were not adjusted for inflation which needs to be considered when reviewing this data.”

This may be an artifact of the submission process, but the appendices are listed in duplicate (p24-28). Apologies – this was an error in the submission process which has been corrected.

Summary

This is a well-written article that contributes to a key gap in the health systems and global surgery literature. I am impressed by this work, and my comments and suggested revisions are all minor. As such, I recommend this article for publication with minor revisions.

Thank you.

Reviewer: 5

Reviewer Name: Lina Roa

Institution and Country: Department of Obstetrics & Gynecology, University of Alberta, Edmonton, Canada

Please state any competing interests or state 'None declared': None

Please leave your comments for the authors below

Thank you for the opportunity to review this manuscript. This manuscript answered an important question on the financial burden associated with surgical care in Sierra Leone using rigorous methodology and addresses the need to move beyond modelled FRP data. Overall, it is a robust manuscript. My main concern is in the exclusion of obstetrics and gynecology surgery, particularly around caesarean sections, since they are one of the most commonly performed procedures, represents a large proportion of the surgical volume in LMICs and often are provided free of charge. The omission of obstetric care requires more explanation and it potentially needs to be recognized as a limitation.

Thank you for your very important comment. Within Sierra Leone tertiary level Obstetric and Gynaecological (O&G) care is provided at a different hospital with the former offered free of charge. Clarification has been included in the study setting section and it has been acknowledged as a limitation to the study. See study setting section: “Obstetric and gynaecological surgical care is delivered at a nearby tertiary referral hospital dedicated to women’s health, where all pregnant and lactating women receive free healthcare under the government’s free health care initiative and therefore not included in this study.”

See limitations section: “Further to this, Sierra Leone tertiary level Obstetric care is provided at a different hospital and offered free of charge. Therefore, costs of accessing this were not included in this study. Further work needs to be done to see if those receiving free maternal healthcare incur any

OOP costs and if informal payments such as tips paid to staff are as prevalent in the obstetric care hospital.”

A few more detailed points are included below.

Abstract

-Methods: CE is defined as >40% of capacity to pay. Elsewhere in the paper the definition used is “>40% non-subsistence expenditure” which in my opinion is clearer. Consider changing the definition in the abstract.

This has been done.

Intro

-Line 23/24 needs clarification. Were the modelling studies in SL? Or in all of West Africa? Or are the authors just saying that SL is in West Africa?

Modelling studies were in SL and this has now been clarified. See: “Modelling studies from Sierra Leone, classed as “least developed” by the UN, and with a population of 7 million reflects these findings; between 84.7% and 49.9% of the population in Sierra Leone is estimated to be at risk of CE if they require surgery.”

-Last paragraph. The metrics were nicely described in a-d. It is not clear if the in-hospital payment mechanism, how costs were met and factors associated were also obtained from the exit survey and if they were also aims of the study or not. Please clarify

This has been re-worded to make it clearer. See: “This study aimed to measure the financial burden associated with receiving surgical care in Sierra Leone by using an exit survey to determine a) direct medical, direct non-medical, and indirect OOP costs to pay for a surgical care episode b) the rate of impoverishment and catastrophic expenditure, c) the wealth characteristics of the population accessing surgical care relative to that of the general Sierra Leonean population, d) the factors associated with higher costs of hospital care, e) the in-hospital payment mechanism (i.e. where and to whom the OOP payments are being made), and f) how costs of accessing surgical care are met, and the factors associated with meeting costs of care”

Methods

-Setting: Please explain why obstetrics & gynecology were excluded given that caesarean sections are one of the most commonly performed surgical procedures. Were these not provided in this hospital?

This has now been addressed. See: “Obstetric and gynaecological surgical care is delivered at a nearby tertiary referral hospital dedicated to women’s health, where all pregnant and lactating women receive free healthcare under the government’s free health care initiative and therefore not included in this study.”

- Data collection: Please clarify if the questionnaire was adapted from existing ones (ref 16-19) or if a whole new questionnaire was developed. If the later, please explain what was better about the new questionnaire developed. Are any of the existing questionnaires validated?

This was a new questionnaire but based on principles and content of existing ones. As there are no validated questionnaires for this specific methodology / study design we feel our approach to developing the questionnaire ensured it was context specific and workable within our study setting.

-Detailed definition of variable is great!

-Is there a reason why for indirect costs only lost wages were considered? Were other costs such as those to caregivers considered?

Yes, only lost wages were considered based on what has been assessed in previous similar studies. Other costs were considered but the research team were also aware of the length of the questionnaire and breadth of information already being gathered and in the interest of avoiding interviewee fatigue we had to make decisions on information that would capture most common reliably reported indirect costs.

-For lost wages what time period was included? Only hospitalization? Or was the recovery time included too?

The time period included was just for the hospitalisation as the interview was performed at the time of discharge from hospital. Therefore, costs associated with / lost wages during the recovery period or further follow-up were not included as this would have been based on unpredictable estimates of future outcomes.

This is acknowledged as a limitation to the study – see paragraph 5 in the limitations section.

- Page 7: How was the conversion from LE to USD done? Was PPP used?

All costs are presented in Le and \$US at the conversion rate of 15th July 2019 (1 Sierra Leonean Leone = 0.00011567 USD). PPP was not used.

Results

-Table 2: Please clarify what is being presented in the brackets after the imputed mean cost

This has been addressed to make it clearer. See table 2: “(\$US (% of subtotal))”

Discussion:

Overall very well written.

-Page 10, line 32-35: If obstetrics and gynecology surgery had been included it is likely that the majority of patients accessing surgical care might not have been males.

We agree – but given that O&G care is delivered at a different hospital, and obstetric care is free, we didn't include this in our survey. Had we included it, it may have been that our results were skewed towards females, given the volume of deliveries.

We have added clarification of this in the discussion see: “The majority of patients accessing surgical care were young males; whether this male predominance is a true reflection of surgical disease burden, beyond obstetrics and gynaecological care, in Sierra Leone or reveals a hidden gender bias in care seeking behaviour is beyond the remit of this study.”

We have acknowledged that O and G was not included in the limitations

See limitations section: “Further to this, Sierra Leone tertiary level Obstetric care is provided at a different hospital and offered free of charge. Therefore, costs of accessing this were not included in this study. Further work needs to be done to see if those receiving free maternal healthcare incur any OOP costs and if informal payments such as tips paid to staff are as prevalent in the obstetric care hospital.”

-Page 11, line 13-14. The Ugandan study referred to include cesarean sections and found that

patients that underwent cesareans were less likely to experience a catastrophic expenditure compared to those who didn't. Is it possible that if you had included cesarean sections in your study, that the risk of CCE and IE would have been less?

This has already been addressed and we acknowledge that this is a limitation to the study. See limitations section: "Further to this, Sierra Leone tertiary level Obstetric care is provided at a different hospital and offered free of charge. Therefore, costs of accessing this were not included in this study. Further work needs to be done to see if those receiving free maternal healthcare incur any OOP costs and if informal payments such as tips paid to staff are as prevalent in the obstetric care hospital."

VERSION 2 – REVIEW

REVIEWER	Leon Bijlmakers Radboud University Medical Centre, Nijmegen - the Netherlands
REVIEW RETURNED	28-Jul-2020

GENERAL COMMENTS	All my earlier questions and suggestions have been addressed; and I believe most if not all of the earlier questions/suggestions from other reviewers as well.
--

REVIEWER	Charles Liu Stanford University Medical Center Stanford, CA, USA
REVIEW RETURNED	30-Jul-2020

GENERAL COMMENTS	The authors have improved on this already interesting manuscript describing an important and much-needed study of the financial burden of surgical care in Sierra Leone. Their inclusion of the full questionnaire/data collection form in the appendix is very helpful. However, several of my previous concerns persist and have not been addressed.  1. Catastrophic and impoverishing expenditures appear in the article title and are the primary outcomes of the study -- but the presentation of these outcomes remains insufficiently clear. In the methods, you state (in your formulas) that if a patient meets /either/ the 40% or the 10% threshold, they are considered to have CE. However, in the results, only the percent of patients meeting the 40% threshold (10%) or the 10% threshold (18%) separately are reported -- not the percent of patients meeting /either/ threshold as was previously defined. (I would expect this to be either exactly 18% or >18%.) Then, in the abstract, only 18% CE is reported. Given this is your primary outcome, I strongly feel you need to choose one of these CE definitions as your main definition and apply it consistently throughout. The other can be reported as a secondary or sensitivity analysis. As it stands, this is unclear and internally inconsistent. 2. I made a typo in my previous review -- if 5% out of 55% of patients not previously impoverished were pushed into poverty by surgical costs, that would be 9%, not 11%. If these numbers are correct, please correct the figure in the abstract. 3. Please define the acronym "OOP" the first time it is used in the manuscript, and separately in the abstract. Currently it is used in the first paragraph of the introduction but not defined until the second paragraph, and it is not defined in the abstract.
---

	4. Please keep percentage results consistent to either 0 or 1 decimal places. Currently there is a mix. Given a sample size of 335 and smaller n for certain subsets, I would recommend 0 decimal places as a more appropriate level of precision. 5. Lastly, the manuscript has increased from 4400 to 4700 words. I realize the authors are responding to a large number of specific reviewer comments, but the length really does hinder readability of the article. I would prefer if it were possible for them to shorten the article (especially the discussion and limitations) and move details into the appendix, as they have already begun to do so. For example, the full questionnaire is included, but the first paragraph of page 6 still lists in great detail all clinical variables collected. This and other methodological details could be provided much more concisely.
--	---

REVIEWER	Carlos Shiraishi-Zapata Hospital EsSalud Talara, Piura-Perú
REVIEW RETURNED	25-Jul-2020

GENERAL COMMENTS	 - On page number 5 authors mentioned the Appendix 6. The appendix table 3 is mentioned in page 9. Also, the appendix table 4 is mentioned in page 12. I suggest that the appendixes are numbered in the order that they are included in the text. - Sample size and power calculation section: Was this calculation performed using a Statistical Software? I suggest to add this information in case so. - On page 22 (Appendix section) at the beginning of the first paragraph, the abbreviation RAs is used. What is the meaning of this abbreviation?
--

REVIEWER	Rolvix Patterson Duke University Hospital
REVIEW RETURNED	01-Aug-2020

GENERAL COMMENTS	All of my comments have been addressed in this revision. As such, I recommend this manuscript for publication.
--

REVIEWER	Lina Roa MD MPH University of Alberta, Edmonton, Canada
REVIEW RETURNED	16-Aug-2020

GENERAL COMMENTS	The authors have satisfactorily addressed all the suggestions and revisions proposed. The manuscript has been strengthened as a result of the revisions and is fit for publication. Thank you for the opportunity to review this manuscript and for your important work on this topic.
--

VERSION 2 – AUTHOR RESPONSE

BMJ Manuscript Response to Reviewers

Reviewer: 3

- On page number 5 authors mentioned the Appendix 6. The appendix table 3 is mentioned in page 9. Also, the appendix table 4 is mentioned in page 12. I suggest that the appendixes are numbered in the order that they are included in the text.

The tables are now numbered in both text and manuscript in the order in which they appear.

- Sample size and power calculation section: Was this calculation performed using a Statistical Software? I suggest to add this information in case so.

This information has been added. See "Sample size was calculated using the USCF online calculator²⁰."

- On page 22 (Appendix section) at the beginning of the first paragraph, the abbreviation RAs is used. What is the meaning of this abbreviation?

This has been changed. See "Research Assistants (RAs) were recruited"

Reviewer: 1

All my earlier questions and suggestions have been addressed; and I believe most if not all of the earlier questions/suggestions from other reviewers as well.

Reviewer: 2

The authors have improved on this already interesting manuscript describing an important and much-needed study of the financial burden of surgical care in Sierra Leone. Their inclusion of the full questionnaire/data collection form in the appendix is very helpful. However, several of my previous concerns persist and have not been addressed.

1. Catastrophic and impoverishing expenditures appear in the article title and are the primary outcomes of the study -- but the presentation of these outcomes remains insufficiently clear. In the methods, you state (in your formulas) that if a patient meets /either/ the 40% or the 10% threshold, they are considered to have CE. However, in the results, only the percent of patients meeting the 40% threshold (10%) or the 10% threshold (18%) separately are reported -- not the percent of patients meeting /either/ threshold as was previously defined. (I would expect this to be either exactly 18% or >18%.) The, in the abstract, only 18% CE is reported. Given this is your primary outcome, I strongly feel you need to choose one of these CE definitions as your main definition and apply it consistently throughout. The other can be reported as a secondary or sensitivity analysis. As it stands, this is unclear and internally inconsistent.

This has been changed. Consistent with the 10% threshold use in the abstract, we have used this as the main measure and presented the 40% threshold results as a sensitivity analysis.

2. I made a typo in my previous review -- if 5% out of 55% of patients not previously impoverished were pushed into poverty by surgical costs, that would be 9%, not 11%. If these numbers are correct, please correct the figure in the abstract.

This has been changed. See “45% of patients were already below the national poverty line prior to admission, and 9% of those who were not below the national poverty line prior to admission were pushed below the poverty line following payment for surgical care.”

3. Please define the acronym "OOP" the first time it is used in the manuscript, and separately in the abstract. Currently it is used in the first paragraph of the introduction but not defined until the second paragraph, and it is not defined in the abstract.

This has been changed. See “Outcome measures: Rates of catastrophic expenditure (CE) (a cost > 10% of annual expenditure), impoverishment (being pushed into, or further into, poverty as a result of surgical care costs), amount of out-of-pocket (OOP) costs, and means used to meet these costs were derived.” “Furthermore 3.7 billion people have been estimated to be at risk of catastrophic expenditure (CE – defined as a total out-of-pocket (OOP) health payment that exceeds a set threshold of the household’s annual income or expenditure) due to a lack of financial risk protection (FRP).^{1,2}”

4. Please keep percentage results consistent to either 0 or 1 decimal places. Currently there is a mix. Given a sample size of 335 and smaller n for certain subsets, I would recommend 0 decimal places as a more appropriate level of precision.

This has been changed throughout the manuscript.

5. Lastly, the manuscript has increased from 4400 to 4700 words. I realize the authors are responding to a large number of specific reviewer comments, but the length really does hinder readability of the article. I would prefer if it were possible for them to shorten the article (especially the discussion and limitations) and move details into the appendix, as they have already begun to do so. For example, the full questionnaire is included, but the first paragraph of page 6 still lists in great detail all clinical variables collected. This and other methodological details could be provided much more concisely.

We are keen that the methods are presented fully and transparently enough for anyone who reads the paper to understand clearly what we have done. We have already shortened the methods to do this, but are concerned that shortening them further will reduce clarity for the many readers who will not read the appendices. We are reluctant to shorten these further. However, we have further shortened the discussion and limitations.

Reviewer: 4

All of my comments have been addressed in this revision. As such, I recommend this manuscript for publication.

Reviewer: 5

The authors have satisfactorily addressed all the suggestions and revisions proposed. The manuscript has been strengthened as a result of the revisions and is fit for publication. Thank you for the opportunity to review this manuscript and for your important work on this topic.